# A Systematic Review of Factors Influencing the Vitality of Public Open Spaces: A Novel Perspective Using Social–Ecological Model (SEM)

**Danning Zhang** [ID]**, Gabriel Hoh Teck Ling** *[ID]**, Siti Hajar binti Misnan** [ID] **and Minglu Fang**

Department of Urban and Regional Planning, Faculty of Built Environment and Surveying,
Universiti Teknologi Malaysia UTM Skudai, Johor 81310, Malaysia
* Correspondence: gabriel.ling@utm.my

**Abstract:** A number of studies address the spatial planning, architectural design, and management of public open spaces (POSs) to curb the overuse of spaces to create high-quality spaces. Little attention has been paid to the problem of underutilization of POSs. Therefore, this paper undertakes a comprehensive analysis of the literature on the factors that influence the vitality of POSs, proposing Bronfenbrenner's social–ecological model (SEM) as a conceptual framework. In this work, we conducted a systematic literature search using the PRISMA method to screen and select articles from three major databases (Science Web, Elsevier, and Scopus). Thirty-four journal articles from 2000 to 2022 were selected for the final review. This study systematically identifies and classifies a set of variables related to the vitality of POSs and develops an SEM-based framework of factors that influence the vitality of POSs. The framework examines the influence of individual user characteristics, the social environment, the physical environment, and the political environment on the vitality of POSs. Finally, strategies to improve the vitality of POSs are proposed: (1) POSs' optimization and promotion strategies should be developed gradually, starting from the most basic needs, stage by stage; (2) To improve the vitality of POSs, we should consider both the general public and special groups; (3) Through the synergistic effect between social, material, and policy environments, the comprehensive improvement of POSs' vitality is achieved. This study provides the latest insights into the vitality of POSs and makes a theoretical contribution by conceptualizing the SEM framework and summarizing the influencing factors at different levels. The study of these factors should also have practical implications, as the results will ultimately provide improvement strategies to help policy-makers and local communities to effectively and sustainably improve the vitality of POSs.

**Keywords:** public open spaces; vitality; users' behavior; social–ecological model

## 1. Introduction

The Sustainable Development Goal calls for providing safe, inclusive, and accessible green and public open spaces (POSs) for all ages by 2030 [1]. POSs are defined as open to all people, including public assembly places and connecting spaces, such as squares, parks, sidewalks, and streets. These spaces are not limited by factors such as gender, race, age, or socioeconomic level [2]. High-quality POSs plays an important role in promoting individual well-being and provide positive value to society, the economy, and the environment of our cities [3]. In the literature review of POSs, it was found that POSs have the problems of lack of attractiveness, low usage, prosperity in the early stage, and abandonment in the later stage in many areas [4–6]. In order to improve user satisfaction and eventually achieve the sustainable development of POSs, the value of POSs, the criteria for good POSs, the quality characteristics of POSs and their evaluation, as well as the management of POSs have been recognized and researched for some time [7–10]. A number of studies have focused on the spatial planning, architectural design, and management of POSs to curb the overuse of

spaces in order to create high-quality spaces. Little attention has been paid to the problem of underutilization of POSs [11].

POSs carry the process of participation in public activities and the expression of the collective will, and people are the ultimate service recipients of POSs. Therefore, the essential measure of spatial quality is people's feelings [12]. The number of users can represent a significant indicator of the performance of a POS. The greater the number and type of visitors, the better the quality of POS, and the more the POS can meet the needs of different user groups [13]. In addition, the diversity of activities, the intensity of space use, and the length of stay of the visitors are good POS indicators [14,15]. In short, a good POS can be performed by different characteristics of users in a variety of activities over different periods of time [16–18]. The lack of interaction in POSs shows that the space cannot satisfy people's needs [7].

Vitality is one of the expressive dimensions of urban design and urban life [19]. Xiangyu [20] argues that users are the main vitality body and that users' activities express vitality. The vitality of a POS is the extent to which the function of the place supports human needs [17] and the ability to provide a decent existence for citizens [21] . Vitality is a product of the visual quality of the environment and the variety of activities it supports [22]. Overall, the vitality of POSs is the quality that makes POSs attractive and usable for continuous activities throughout the day [23,24]. Moreover, the vitality characteristics of POSs are usually expressed by the duration of the population's activities, the density of the population, and the diversity of the different activities [25]. In summary, user behaviors (such as access, stay, and use) determine the vitality of POSs. A systematic study of the factors affecting the vitality of POSs is of great significance for promoting the sustainable development of POSs and improving public satisfaction.

The social–ecological model (SEM) is a framework that has been widely used in public health research and practice and is widely accepted in behavioral research [26,27]. The model integrates the various levels of behavioral influence to paint a clear and comprehensive picture of the factors that influence behavior and ties behavioral activity into a broader context, favoring a broad, multidimensional view of the factors that influence behavioral activity [28]. Several multilevel models specific to physical activity have been proposed that incorporate individual, social, physical, environmental, and political variables [29,30]. In this model, factors affecting behavior are represented in layers, and each layer affects the next layer, and personal behavior change is also affected by each layer [31]. Elder et al. [32] suggested that the environment affects individual behavior, and behavior also affects the environment. Changes in factors external to the individual, such as social circumstances, policies, and environmental factors, increase the likelihood that behavior change will occur [33].

The individual level is the focus of the model, including individual characteristics such as age, gender, knowledge, income, health status, and attitudes that lead to increased or decreased likelihood of performing a behavior [34]. The social environment, including cultural and interpersonal interactions, significantly influences physical activity behavior, including active and passive activities [35]. For example, having an active classmate, colleague, or family member can have a positive impact on your behavior. The physical environment includes natural and artificial environments that facilitate behavioral activities, such as facilities, places, and landscapes [36]. The political environment refers to the rules, regulations, and policies that influence behavior [37].

Based on the social–ecological theory and the systems theory, SEM systematically analyzes the influencing factors of behavioral activities and then clarifies the comprehensive influence of the individual level, physical environment, social environment, and political environment on the multi-level factors of behavioral activities. This model provides a comprehensive analytical perspective and logical framework to accurately understand the influence mechanism of behavioral activities among POS users and to implement an effective intervention. Therefore, in reviewing the literature on influencing the vitality of POSs, this paper uses the SEM as an organizational framework to summarize the factors

that influence the vitality of POSs. In order to improve the vitality of POSs, it is important to understand the factors influencing user behavior and the mechanisms of action of these factors on the vitality of POSs.

## 2. Methods

This systematic review followed the Preferred Reporting Items for Systematic Review Recommendations (PRISMA) guidelines protocol and the construction of systematic reviews in research guidelines, and the PRISMA method, a systematic literature review method, was used for the literature screening and review, which is divided into four phases, namely identification, screening, eligibility, and inclusion [38–40].

In the identification phase, three databases (Web of Science, Scopus, Elsevier (Science Direction)) were searched for articles using a search string created with Boolean operations ("public space" OR "green space" OR "open space") AND ("vitality" OR "visit" OR "use") AND ("influence" OR "impact" OR "factor"). The search was limited to articles with research reports published in English between 1 January 2000 and 29 September 2022. Two reviewers (Zhang and Fang) independently searched for articles, reviewed the titles and abstracts of the articles found, reviewed the full texts, and selected articles for inclusion. In addition, the articles included in this review had to meet the inclusion and exclusion criteria listed in Table 1.

**Table 1.** Eligibility criteria.

| **Articles Inclusion Criteria:** |
| --- |
| Studies of English writing<br>Publish time: 2000–2022<br>Scholarly papers<br>The studies had to present original peer-reviewed research providing quantitative information about the relationship between the vitality of public open spaces and users' individual, the physical environment, and social environment and policy factors. |
| **Exclusion criteria:** |
| Papers that are duplicated within the search documents<br>Papers that are not accessible, review papers<br>Papers that are not primary/original research |

All search articles were retrieved and uploaded in the Zotero tool for systematic reviews. The abstract and title of the study, as well as the full text screening were conducted by two independent reviewers (Zhang, Fang), with a third reviewer used to resolve disagreements.

For data extraction, a standard template was used, containing details of each articles' title, author, date, title, context, sample size, data collection, influencing factors, and outcome of using POSs.

Quality appraisal of studies was conducted by two reviewers using the QUALSYST, as QUALSYST can assess the quality and potential bias in a wide range of study designs, from experimental to observational [41]. All articles were evaluated in five areas: study context, data collection methods, sample size, influencing factors, and results.

According to the above search method and inclusion criteria, a total of 970 papers were found in the first step, including 301 in Web of Science, 442 in Scopus, and 227 in Elsevier. Then, we removed 239 duplicates and 146 other irrelevant articles in the screening process. We then reviewed the titles and abstracts of the 585 articles to determine whether these articles contained research that focused on commons in terms of the relationship between the vitality of POSs and individual users, the physical environment, the social environment, and political factors. In the next step, we further evaluated the 66 articles by carefully reading their full text. In doing so, we used the research content, background, terminology interpretation, etc., to determine whether the articles selected in the second step met the inclusion criteria of the references. Finally, 34 articles from 2000 to 2022 were

selected for review. Figure 1 shows the screening process using the PRISMA method for the eligible articles.

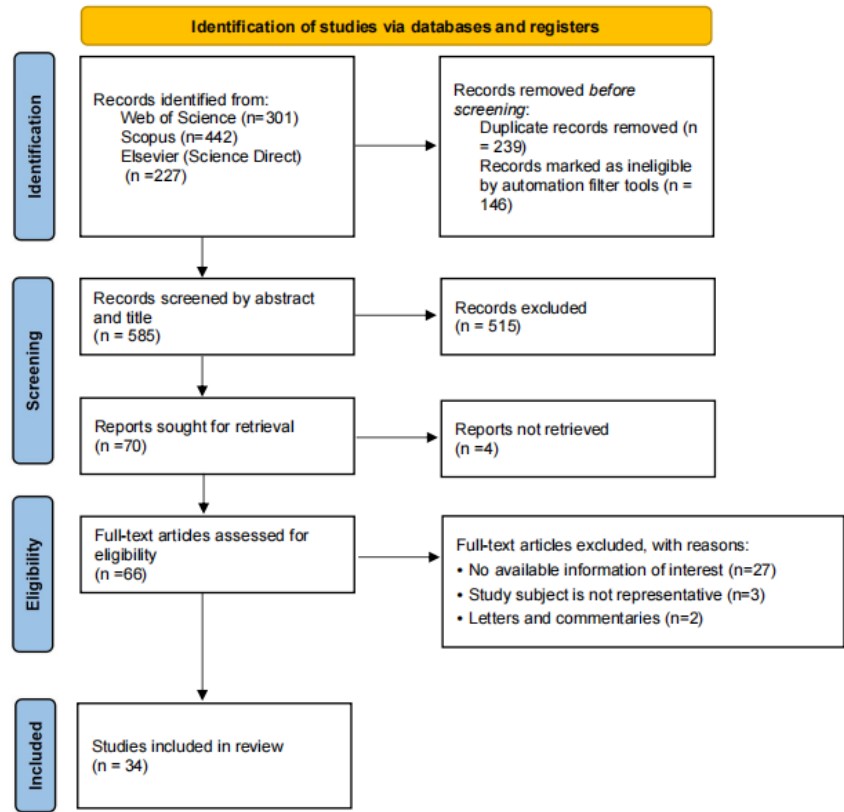

**Figure 1.** The systematic literature review process was captured via a PRISMA flowchart.

In reviewing the literature on the influencing factors for the vitality of the POSs, this paper also used SEM as an organizational framework to synthesize studies related to the user behavior impact of POSs. The factors affecting the vitality of POSs were determined through literature combing. Combined with SEM, the influencing factors were described in different levels, and the factors were summarized into four levels of SEM, which were individual characteristics, social environment, physical environment, and policy environment. Following an overview of the literature, we will discuss the understudy of the current literature, the possible development of SEM in the study of influencing factors, and strategies for the POSs' vitality boost.

## 3. Results and Discussion

Using a standard template that was used for data extraction, the information and results of these 34 reviewed articles are summarized in Table 2. It provides details on the title, author, date, title, context, sample size, data collection, influencing factors, outcome of using POSs, and SEM perspective for each article. Of the 34 articles included in this review, the largest number of articles examined the physical environment of POSs, with 31 articles examining the factors that influence the vitality of POSs from the perspective of improving spatial quality. 20 articles address user characteristics, including gender, age, income, and educational background, illustrating the influence of individual characteristics on the use of POSs. 10 articles examined the influence of social environment factors on the vitality of POSs. 5 articles examined the impact of policies on the vitality of POSs, focusing on impacts related to planning supports and behavioral interventions. The following article addresses each of the influencing factors.

**Table 2.** Characteristics of included studies.

| Study (Authors and Year) | Context | Sample Size and Data Collection | Influencing Factors | Results of POS Use | SEM Perspective |
|---|---|---|---|---|---|
| Fan et al., 2021 [42] | Urban | 178 POS Baidu heat map data Geographical detector | Park size, landscape shape index, park facilities, water size, vegetation coverage, road density, traffic convenience, distance from the urban center, park-surrounding facilities | Facilities around are the most significant drivers of park use, and there is a bi-variate enhancement or non-linear enhancement of the interaction effect manifested between each pair of drivers. | Physical Environment Social Environment |
| Siderelis et al., 2012 [43] | Urban, Periphery, Rural | 1350 POS Annual Information Exchange (AIX) | Numbers of workers, facilities of park-lands, capital investments. | Increased labor to maintain parkland increases park attendance, but more capital is noted, and the number of facilities does not increase park utilization. | Physical Environment Political Environment |
| Aliyas, 2020 [44] | Urban | 40 People Semi-Structured Interviews. | Characteristics of parks (e.g., size, facilities, accessibility, natural landscape, and safety)/Social factors (e.g., bad behavior and social support)/Individual factors (e.g., time, weather, negative attitudes, and health condition). | Physical characteristics of parks, social factors, and personal factors influence the selection of parks for physical activity. In addition, the combination of these factors influences the selection of physical activity parks for all age groups. | Individual Characteristics Social Environment Physical Environment |
| Yuan et al., 2021 [45] | Urban | 441 POS Field surveys | Transportation, built environment, population, the density of residential entrances and exits, walkable areas, and density of retail and service facilities. | The vitality of a street is related to the density of the population in the environment, the density of the residential entrances and exits, the proportion of walkable areas, and the density of retail and service facilities. | Physical Environment. |
| Yu et al., 2022 [46] | Urban | 1200 People GPS Path Tracking Geographical detector | Percentage of POS area, accessibility, population, percentage of commercial land, an area occupied per capita, percentage of residential land, and transportation convenience. | The percentage of public open-space area, accessibility, population density, percentage of commercial land use, and per capita occupancy influence the intensity of public open space use. | Physical Environment |
| Rivera et al., 2021 [47] | N/A | 34 People Interviews | Park's natural features, sports facilities, aesthetics, location, green spaces, barbecue areas, seating, organized activities, shelters, safety, and social factors. | The park's natural features, sports facilities, aesthetics, safety, social factors, and location will promote park visitation. Sports facilities and green spaces will promote physical activities. Furthermore, barbecue areas, sports features, seating, organized activities, and shelters will encourage socialization. | Physical Environment Social Environment |
| Medina et al., 2022 [48] | Urban | 944 POS Field surveys | Food and wellness environment, maintenance, amenities, legibility, security, perceived environment, and urban environment. | Food and wellness environment, maintenance, amenities, legibility, security, and environment will impact participants' attendance in public open spaces. | Physical Environment Political Environment |
| Fongar et al., 2019 [49] | Urban Suburban Rural | 1010 People Questionnaire | Age, gender, the degree of urbanization, households with children under 18 years of age, perceived quality, distance, education, and noise. | Norwegians perceive their green spaces to be of good quality, and the perception of higher quality has a positive impact on green space visitation. | Individual Characteristics Physical Environment |

**Table 2.** *Cont*.

| Study (Authors and Year) | Context | Sample Size and Data Collection | Influencing Factors | Results of POS Use | SEM Perspective |
|---|---|---|---|---|---|
| Wang et al., 2022 [50] | Urban | 238 POS Map data | Accessibility, surrounding facilities, integration, number, area, public transport, facility type and quality, landscape, maintenance, and quietness. | The quality of open space has a more significant impact on the number of elderly visitors than accessibility factors. | Physical Environment Political Environment |
| Chen et al., 2018 [51] | Urban | 686 POS GPS data | Accessibility, surrounding facilities, POS integration, POS number and area, public transport, facilities, landscape, maintenance, and quietness. | The quality of an open space has a more significant impact on the number of elderly visitors than accessibility factors. | Physical Environment Political Environment |
| Wan et al., 2015 [52] | Urban | 263 People Questionnaire | Facilities, perceived naturalness, accessibility, attitude, usefulness, subjective norm, behavioral control, behavioral intention, behavior. | Perceived provision of facilities, accessibility, attitude, subjective norms, PBC, behavioral intention, and usefulness relate positively to the behavioral intention to use urban green spaces. | Physical Environment Individual Characteristics |
| Jiang et al., 2018 [53] | Urban | 91 People Questionnaire | Age, time, walking system, rest space, square landscape and facilities design, seasons, social interaction, media art. | Comfort, diversity, public activities, and social interaction will enhance the vitality of public spaces. | Individual Characteristics Physical Environment Social Environment |
| Ye & Qiu, 2021 [54] | Urban | 70 POS Social-media data Message Board | Park age, square, facilities, water, accessibility, park attributes, distance to the city center, accessible area for a 10 min trip, surrounding point of interest, and population. | Large areas, adequate facilities, reasonable layout, accessibility, recreational facilities, services near points of interest, and prosperous organization of activities contribute to the improvement of park utilization. | Physical Environment Social Environment |
| He et al., 2022 [55] | Urban | 56 POS Gaode Maps Google Images | Age, gender, park attributes, park vegetation, surrounding, and serviceability. | Gender, age and park aggregation influence users' participation in parks. | Individual Characteristics Physical Environment |
| Addas, 2022 [56] | Urban | 409 People Questionnaire | Gender, nationality, educational level, occupation, socio-demographic attributes, seasonal variation, the pattern of use, accessibility, benefits or purpose of park use, park attributes, and policy. | Urban parks are mainly used for spending time with others, followed by mental relaxation, sports activities, and children's company. Seasonal changes, socio-demographic attributes, and urban management largely affect the park's use. | Individual Characteristics Physical Environment Political Environment |
| Kaymaz et al., 2019 [57] | Urban | 1299 People Questionnaire | Age, gender, property type, outdoor activity preferences, neighborhood's built environment, accessibility, safety, space design, temperature, density vegetation, shady areas, facilities, duration of living in the neighborhood. | The benefits of outdoor activities, safety concerns, and design features are the three main factors that influence the use of green space by parents and children. Social, cultural, and physical environments are all influences on children's green space use. | Individual Characteristics Social Environment Physical Environment |

**Table 2.** *Cont.*

| Study (Authors and Year) | Context | Sample Size and Data Collection | Influencing Factors | Results of POS Use | SEM Perspective |
|---|---|---|---|---|---|
| Schipperijn et al., 2010 [58] | N/A | 11,238 People Questionnaire | Distance, accommodation type, size of the municipality, ethnic background, reasons for visiting green space. | Enjoying the weather and fresh air were the mainly reasons for respondents to visit green spaces. For most Danes, distance to green spaces is not a limiting factor for visiting green spaces. | Physical Environment Individual Characteristics |
| Van Hecke et al., 2018 [59] | Urban | 173 People Questionnaire GPS Interview | Education, ethnicity, location, gender, age, sports club membership | The purpose of visiting public open spaces was recreational, and participants spent more time when accompanied. Boys and less-educated adolescents were more likely to use public spaces. | Individual Characteristics |
| Liang et al., 2022 [60] | Urban | 402 People Questionnaire | Exercises, safety, accessibility, social interaction, consumption, public participation, environment, age, income, education, policy intensity, the effectiveness of a policy, and gender. | The primary needs of residents for green space are environmental connection, safety, and accessibility, and the needs vary by gender, education level, and income. | Individual Characteristics Social Environment Physical Environment Political Environment |
| Burrows et al., 2018 [61] | Urban | 865 People Questionnaire | Gender, age, visit the park alone or with others, proximity of residents to the park, quiet, visit reason. | When and why individuals go to the park, the distance from the park has a more significant impact on the visit frequency than age, gender, and impressions of park sound levels. | Individual Characteristics Social Environment |
| Zhang et al., 2013 [62] | Rural | 364 People Site survey Questionnaire | Gender, age, education, occupation, income, household size, residence, house size, dwelling location, vegetation, topography, garden ornaments, historicity, and recreation facility. Safety, naturalness, uniqueness. | Differences in age, gender, income, and education level determine the demand for green space recreation. Green space environment and accessibility are the two main factors that influence residents' choices. | Individual Characteristics Physical Environment |
| Vidal et al., 2022 [63] | Urban | 979 People Observation | Age, physical activity level, status, mobility, weather, day period, temperature, size, deprivation cluster, socio-economic profile, space shape, vegetation, urban furniture, and surroundings. | The use of urban public green space is related to the situation of the users, the poverty level of the surrounding environment, and the design of the space. | Individual Characteristics Physical Environment |
| Misiune et al., 2021 [64] | Urban | 444 People Questionnaire | Ecosystem services, nature benefits, distance, recreational infrastructure, safety concerns, noise, vegetation allergies, accessibility, free time. | The most important factors that attract people to green spaces include distance and safety, leisurely walks, enjoyment of fresh air, observation of nature, relaxation, and recreation. | Individual Characteristics Physical Environment |
| Zhang et al., 2018 [65] | Urban | 127 POS Social-media data | Park size, entrance fee, presence of water, vegetation cover rate, number of bus stops, population density, average housing price, number of nearby parks, distance to urban center. | The number of bus stops is positively correlated with park visitation. Improving park accessibility through public transportation and planning small, accessible green spaces are effective in improving park use. | Physical Environment Social Environment |

**Table 2.** *Cont.*

| Study (Authors and Year) | Context | Sample Size and Data Collection | Influencing Factors | Results of POS Use | SEM Perspective |
|---|---|---|---|---|---|
| Sanesi et al., 2006 [66] | Urban | 351 People Questionnaire | Age, sex, function, size, maintenance and structures, facilities, safety, marital status, area of residence. | The use of public green space is closely related to age, gender, marital status, and area of residence. | Individual Characteristics |
| Mu et al., 2021 [67] | Urban | 150 People Field investigations Observations Questionnaire | Age, congregation spaces, locations, facilities, type of land use, accessibility, diversity of sub-spaces and activities, plants, landscape, water quality. | Park spatial vitality varies across time and space, and spatial vitality is influenced by park location, amenities, and visitors' age group. | Individual Characteristics Physical Environment |
| Wang et al., 2021 [68] | Urban | 634 People Questionnaire | Age, reasons for SUGS use, socio-demographic factors, personal factors, spatial attributes of residence, park features factors, with a child under seven years of age, noise, facility, income, distance, and residential green spaces. | Relaxation and rest, physical exercise, and meeting friends were the most common reasons for using SUGS. Age, willingness to access nature, having children under seven years of age, noise, and facilities were positively associated with SUGS use. Income, distance from home to park, and residential green space were negative. | Individual Characteristics Physical Environment |
| Kiplagat et al., 2022 [69] | Urban | 1030 People Observations Questionnaire | Size, accessibility, maintenance, seats, security, vegetation, activities, facilities, parking lots, distance and cost to green spaces, socio-economic attributes of users (gender, age, marital status, occupation, household size, income, education level). | Green spaces that exhibited the most attributes were heavily visited. Gender, marital status, and educational attainment were significant socio-economic predictors of greens pace use. | Individual Characteristics Physical Environment |
| Aziz et al., 2020 [70] | Urban | 356 People Questionnaire | Age, ethnicity, gender, intention, self-efficacy, health condition, activities, features, routes, characters, distance, size, attractiveness, accessibility, comfort, and safety, family, peers, professionals, and community. | The majority of users of this park are Malays, with a higher number of people in the 26–32 age group. People within a 2 km radius visit this recreational area more often to rejuvenate and escape the busy city life. | Individual Characteristics Physical Environment |
| Zhou et al., 2022 [71] | Urban | 54 POS Geospatial data | Park area, distance to SBD, seats, recreational facilities, surroundings (toilets, retail shops, restaurants, bus stops, area of comprehensive parks, area of community parks, density of the traffic roads). | Socio-economic features of surroundings (population, housing price) Higher residential populations, more public restrooms, and larger open spaces are more likely to support small park access. However, the distance from downtown, surrounding large parks, and major roads do not support remote park access. | Physical Environment |
| Pratiwi1 et al., 2022 [72] | Urban | 105 People Observations Questionnaire | Age, gender, education, monthly revenue, accessibility, physical elements (pedestrian way, street furniture, visual along the corridor, social space). | Vision, ambiance, and spaciousness are considered to be the main attractions of heritage public spaces. | Individual Characteristics Physical Environment |

**Table 2.** *Cont.*

| Study (Authors and Year) | Context | Sample Size and Data Collection | Influencing Factors | Results of POS Use | SEM Perspective |
|---|---|---|---|---|---|
| Zhu et al., 2020 [73] | Urban | 90 POS Social-media data Gaode Map | Entrance fee, vegetation, water in the park, the density of facilities, distance to the urban center, population density, density of bus stops, diversity outside the park, and urban function density outside the park. | The vitality of urban parks decreases along an urban–rural gradient. Water in parks, the density of facilities, and nearby population density had a significant positive effect on park vitality. | Physical Environment |
| Chen et al., 2016 [74] | Urban | 112 POS Interview Environment scan Observation | Accessible lawn area, woodland area, footpath length, pavement, facilities, commercial facility sites, seats, shading devices, parking facilities, trash cans, landscape, and lighting. | Large accessible lawns, well-maintained sidewalks, seating, commercial facilities, and water features are essential features that can increase the use of open space in a community. | Physical Environment |
| Liu et al., 2021 [75] | Urban | 102 POS Baidu heat map data Geographical detector | Site design characteristics (shorelines open degree, public service facility density, non-motorized vehicle lane density) Traffic accessibility (bus station coverage index, road network density, non-motorized vehicle lane accessibility) Surroundings building and population. Service facility (surrounding commercial service density, catering facilities) | Site design, surrounding population, and services have a significant positive impact, while accessibility has a negative impact on vitality. | Physical Environment Social Environment |

### 3.1. Individual Characteristics

In the SEM, the focus is on the individual trait dimension that increases or decreases the probability of performing the behavior [31]. Furthermore, Bronfenbrenner has identified three individual-level traits that influence an individual's development: demand, resources, and force. Demand depends on a person's social role, which is determined by gender and age. Resources refer to non-material characteristics such as intelligence, skills, abilities, and experience, as well as material characteristics such as income and education. In addition, force refers to psychological factors such as motivation and emotion [76,77]. Psychological factors map the external environment at the psychological level of the individual and are key to behavior change. The complexity of human behavior mainly reflects how external factors influence psychological factors and behavior change. Behavioral attitude and behavior motivation are mainly psychological factors that influence activity behavior [78,79].

A literature review of factors affecting the vitality of POSs found that age, gender, and education level are the most common individual characteristics. Age is a critical factor influencing user behavior toward POSs, and there are evident differences between age groups in accessing and using POSs [62,66]. More specifically, seniors and youth prefer neighborhood parks and young people are the main users of urban green spaces outside high-rise buildings and office areas [55]. There are differences in the amount of time people of different ages spend using POSs [53]. As seniors have more free time, they visit urban green spaces more frequently [63]. Addas [56] found that park use and accessibility varies by age, with visitors in their twenties and forties typically accompanied by a partner, child, or parent when visiting POSs. Older people have a higher need for green space than younger people, especially in terms of exercise, accessibility, social interaction, and connections to the environment [60].

Gender as another influencing factor was inextricably linked to age group. Among the respondents under 45 years old, men have a greater need for recreation in POSs than

women, and women over 45 have a greater need than men [62]. In the study on the variability of park visits, He et al. [55] found no difference in the average number of daily park visits among adolescents and teenagers by gender. However, in the older age group, the average number of daily visits was almost twice as high for older females as for males. Van Hecke et al. [59] indicated that male adolescents are more likely to use POSs and gender has a significant influence. Liang et al. [60] found that the functional needs of males and females differ significantly, with males having a lower preference for spaces and females having a higher preference for spaces, and females in particular preferring POSs, which fulfills the functions of public participation, consumption, and environmental reference. Men prefer city parks and women prefer neighborhood parks. In addition, Sanesi & Chiarello [66] reported that men use POSs primarily for sports, while women prefer facilities with play opportunities for children.

Educational factors may also affect an individual's participation in POS behavior. Liang et al.[60] found that residents with higher education placed greater importance on the need for safety, accessibility, and environmental connection in POSs, while residents with lower education placed greater importance on interaction, public participation, and consumption needs. In their survey, Van Hecke et al. [59] found that a POS was eight times more likely to be used by those with technical education than those with general education. Resident demand for POS recreation decreases as education level increases [62]. However, the frequency of visits to POSs was higher among residents with high levels of education than those with low levels of education [49,56,69].

In addition to the above factors, income, marital status, and physical condition are also factors that influence the behavior of POS residents. Liang et al. [60] argued that different incomes have different needs in terms of POSs. More exactly, compared with the middle and high-income groups, the low-income groups significantly exhibit more needs of sports, social interaction, public participation, and consumption. High-income groups relatively prefer the safety and accessibility. P. Wang et al. [68] found that low-income residents used POSs more frequently than higher-income residents. Specifically, for residents with monthly incomes below USD 960, the need for POSs increases with income, while the need for POSs decreases with income when monthly income exceeds USD 960 [62]. Marital status also affects the usage patterns of POS residents [56,66,69]. Marriage or partnership can be considered a symbol of love and companionship. When divorced, separated, or widowed, it is natural to assume that these people become lonely and unhappy. Therefore, green space, as a place to help reduce loneliness and improve social cohesion, will naturally be attractive to these people and thus become important. Moreover, the health status of residents affects the utilization behavior of POSs [70]. A person's physical condition can affect the feasibility of their travel, and the purpose of using POSs. For example, if a person physically needs walking exercise, he is more motivated to go to the POS.

However, the analysis revealed that need, resources, and force interact with and influence each other in the individual characteristics and that looking at a single factor does not have significance for the analysis results. For example, the effect of gender alone is not significant, but after differentiating age groups, the effect of gender is significant. A person's demands, resources, and forces will influence the user's behavior in POSs. A person's social role will change their attitude and behavior motivation. Their wealth, wisdom, and knowledge will also change their attitude and behavior motivation. Therefore, a combined analysis of the factors is required when analyzing the influencing factors at the level of individual characteristics.

### 3.2. Social Environment

The social environment can reflect the interpersonal relationship or cultural atmosphere in a specific place and will affect the specific behavior of POS users. Behavior-related theories often incorporate social and environmental factors into constructing behavioral decision-making relationships. This literature review summarizes the factors of social environment into the following two points: social support and the sense of belonging.

Social support can be defined as the resources gained through communicative interactions with others, often in the form of emotional, informational, and material support. For example, companionship, encouragement from friends or family, advice from professionals, etc., influence POS use behavior [80]. This is because people are more likely to engage in physical behavior when supported or encouraged by their social environment [29]. Family or friend companionship is the primary motivation for visiting a POS or green space [44,56,58,61]. Peer influence can be a factor in encouraging a preference for POSs [47]. Parental attitudes can restrict children's behavior in POSs, and taking children outdoors is the only incentive for adults to visit green spaces [57].

An environment with a sense of belonging will attract people who identify with this atmosphere experience to visit repeatedly, which will further strengthen the belonging atmosphere of the place. The orderly environment of society also affects the use of POSs. Specifically, Aliyas [44] found in the interview that respondents were biased toward going to parks with many people or to POSs during crowded hours, parks with homeless people or drug users sleeping, where they would feel less safe. Racial differences can also affect access to POSs, with people of different races visiting POSs less frequently than locals [58,59].

Therefore, in the SEM, the social environment requires special attention in shaping interventions on POS behavior, as they are vital in determining and motivating physical behavior in POSs [44,49,51].

### 3.3. Physical Environment

Among the studies on factors influencing the use of POSs, the quality of the physical environment is the most mature, with 31 of the 34 literature reviews in this paper researching the impact of the physical environment of POSs. The physical environment can be a barrier or facilitator of physical behavior engagement. Of all the physical environment factors, accessibility is the most critical factor affecting users' access to and use of POSs, such as distance from home, distance to downtown, public transportation connectivity, etc. Distance is negatively correlated with use. The greater the distance, the lower the likelihood of use [42,44,47,50,67,70]. Several studies have suggested that the primary users of POSs are those who live within 300 m of the POS [49,58,68]. However, Liu et al. [75] pointed out that for waterfront open spaces, accessibility can have a negative impact on spatial vitality because if the POS level of safety and management is low, high accessibility may harm POS use, such as the problem of uncontrolled parking.

Safety is also an important attraction factor for POSs. Users who feel unsafe when using POSs will reduce visiting behavior and dwell time, which will reduce the vitality of POSs [44,57,62,64]. In addition, the facilities and functions of POSs have a positive effect on the vitality, such as children's facilities, recreational facilities, walking paths, barrier-free facilities, seating, etc. [42,46,47,57,64,74]. Interestingly, the number of toilets also affects the vitality of POSs to some extent [67,71]. Another important attractive attribute of POSs is the natural landscape and plants [44,49,51]. More specifically, the researchers found that adding water features increased the number of visitors and length of stay in POSs [65,73,74].

In addition, the maintenance and management of POSs will also encourage users to use POSs [42,70] . Organizing various activities and establishing rules for using POSs are effective ways to improve the vitality of the space [53,57,67]. In general, the quality of the physical environment of POSs is positively correlated with POS vitality within a specific range.

As Maslow [81] described in their influential theory of human motivation, people are motivated by various needs. These demands are "organized into a hierarchy of potential." Some needs are more fundamental than others. One must meet these more basic needs before one can consider higher levels of needs. However, in the articles describing the influencing factors of the physical environment, few articles suggest that those influencing factors are the most urgent or basic needs of POS users, and blindly improving the physical environment may not achieve the desired effect. For example, when Aliyas [44] investigated park use, interviewers would instead choose a park farther away than a less lit park near

their home because it makes them feel unsafe. Therefore, the subsequent research should focus on the stratification of user needs.

### 3.4. Political Environment

The policy is a higher-level process that substantially impacts the lower tiers in SEM [82]. According to these collected articles, research on the policy environment factors for the vitality of POSs has focused on planning support. Siderelis et al. [43] stated that to increase POS attendance, the government needs to increase the workforce to maintain public lands. When a POS is already utilized to a certain extent, adding more facilities or increasing investment will not increase attendance. However, public land preservation is significantly associated with increased utilization.

Regarding the right to equity in POSs, the planning of POSs should focus on social equity, especially considering the needs of disadvantaged groups, such as the elderly, to increase the visits to POSs [50]. Addas [56] argued that the way to improve the use of POSs is through effective community management, with the government creating new green spaces, maintaining existing green spaces, and increasing the amount of green space per capita based on visitor perceptions and preferences. Liang et al. [60] found that "15-minute community planning" (locating green spaces within 15 min of where people live) was the best policy for increasing green space use by testing three policy scenarios. Meanwhile, he found that policies for social interaction, commerce, and public participation were sensitive factors for visitor access to early POSs. In contrast, policies for safety and security, environmental quality, and accessibility were sensitive factors in the later stages of POS use. In addition, proper mixed functional planning and dynamic zoning of POSs also positively impacted the improvement of vitality [73,75]. The limited research on the policy environment affecting the vitality of POSs suggests that a variety of policies interfere with the vitality of POSs and that adequate planning policy support can enhance the vitality of POSs.

Through an extensive interpretation of the SEM and a literature review of the four dimensions of individual characteristics, social environment, physical environment, and political environment, the key factors influencing users' behavior related to the vitality of POSs are summarized as shown in Figure 2.

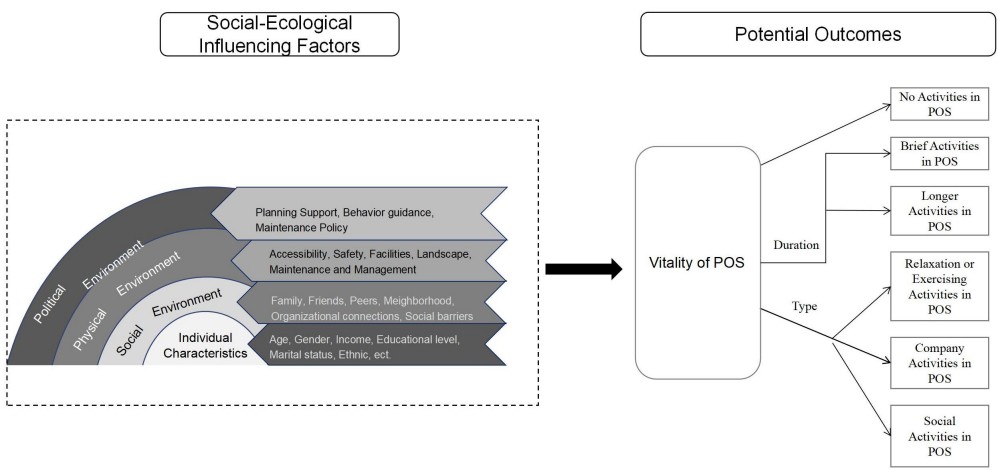

**Figure 2.** Elaboration of the SEM of determinants of the vitality of POS.

### 3.5. Limitation of This Literature Review

Villages and cities differ in lifestyle, social environment, history, and culture. Rural POSs have mixed functions, such as a drying dam, usually used as a party or sports leisure space; people will cool or fish by the pond. The utilization of POSs in rural areas is not only for recreation but also for the needs of production and life [83]. This can also explain why the traditional rural POSs have high vitality, while the new POSs loses the traditional

emotional bond and low vitality. However, most of the studies selected in this review were from urban areas, and only two articles discussed POSs in rural areas. Therefore, the limitation of this review is the inability to make general statements about the factors influencing the vitality of POSs in rural and urban areas. In future studies, historical and social development in rural and remote areas need to be considered to explore the dynamic impact of POSs. Nevertheless, the large sample number of the articles selected in this review could complete the vitality analysis of POSs in urban areas.

## 4. Conclusions and Recommendations

The broad interpretation of SEM has introduced vital factors that influence users' behavioral activities at POSs. However, several arguments in the above mainstream literature must be critiqued for developing a more robust and hierarchical approach to examine the reasons influencing users in conducting behavioral activities at POSs.

### 4.1. Neglect the Impact of Policy

The policy provides an organizational structure and guidance for collective and individual behavior. It can be defined as a legislative or regulatory action taken by federal, state, municipal, or local governments, governmental agencies, or non-governmental organizations such as schools or companies. Policies include formal and informal rules and design standards that may be explicit or implied—for example, planning, reporting, or other research findings that express a specific strategic vision. Policies control and influence behavior at many levels, with some having direct effects while others having indirect effects mediated through many other factors [84]. In behavioral interventions, policy factors are usually placed at the top of environmental measures and are the antecedents of the impact of the extrinsic environment on behavior. SEM describes the policy environment as the farthest layer of behavioral effects. The above content fully demonstrates that the policy environment is an important dimension that cannot be ignored in behavioral activity research.

However, through this literature review, in the empirical research with the SEM, it is difficult to systematically explore policy factors, limited by the specific conditions of the research (such as the acquisition of data and the coordination of various departments). Most of the studies demonstrate only some factors. For example, when studying how to improve the use of green space, Liang et al. [60] tested the effects of three green space planning policies, focusing on the impact of the quantity and quality of urban green space provided on the use. Regarding the discussion of policy factors, the relevant research has not yet formed systematic content.

### 4.2. Physical Environment Improvement as a Panacea

Among the articles on improving the vitality of POSs, most of the views focus on improving the physical environment of POSs, and 31 of the 34 articles selected in this study discussed the influence of the physical environment. The researchers have focused on improving the landscape of POSs, increasing the facilities, improving the accessibility of POSs, etc. It seems that the physical improvement of POSs is regarded as a panacea. However, the improvement of the physical environment did not bring the expected results, and instead brought new problems. For example, the new and beautified POS is less attractive than the old POS [85]; an optimized POS prospers in the early stage and declines late [86]; and the disconnect between the supply of POSs and social demand, since the unilateral pursuit of space quantity, scale, and image, ignoring the quality of the space, such as comfort, convenience, and practicality [83].

Significant changes have taken place in the modern lifestyle. Work occupies young people most of the time. According to Statista, the global employee population reached 3.32 billion in 2022, an increase of about 1.04 billion from 2.28 billion in 1991 [87]. Moreover, Internet electronic products occupy much free time, so people reduce their dependence on POSs. Moreover, POSs assumed complex functions in the past, especially in rural areas.

Some POSs are both production spaces and living spaces, so the utilization rate of POSs is high, but now POSs often have a single role and a single function, which reduces the necessity of using POSs [88].

In the process of improving the vitality of POSs, the improvement of the physical environment is very important. However, it is not easy to establish a good POS without considering the characteristics of user groups, social environment, and policy environment.

### 4.3. Future Development of SEM

The advantages of the SEM application are evident. With the gradual maturity and development of the SEM, people have extensively studied the influencing factors of behavioral activities in a multidisciplinary field. For example, psychology took behavioral psychology to explore the relevant interventions [89]. Public health improved health through health education and guidance [90]. In addition, the SEM is generally recognized in behavioral activity research mainly because the model integrates behavioral activity research into a more comprehensive context.

However, the SEM also needs to improve. First of all, the SEM does not elaborate on the logic of the influence of various factors on behavior. Although most related studies have used concentric circle structures to express the circle layer relationship between factors, it is difficult to identify the relative advantages, action relationship, and causal logic between factors. Therefore, based on this model, scholars applied other theories to explain the action relationship between factors. Elder et al. [32] integrated the functional learning theory, the social cognition theory, and the organizational change theory in SEM by conducting the activity trial study of adolescent girls. Alfonzo [91] combined SEM with Maslow's hierarchy of needs to determine the influencing factors influencing residents' walking, then proposed that one must meet these more basic needs before considering higher levels of needs. In terms of influencing factors on the vitality of POSs, the potential influencing factors of the vitality of POSs can be systematically summarized and sorted out through the literature review, based on the support of the SEM. However, although the SEM points to the multi-dimensional interaction between the influencing factors, it does not elaborate on their specific relationship. In addition, the behavioral decision is complex, so it is necessary to establish a theoretical model of the interaction of influencing factors to achieve clear guidance for discussing the influence mechanism of the vitality of the POSs. To this end, it is necessary to combine SEM with behavioral-related theories (Figure 3).

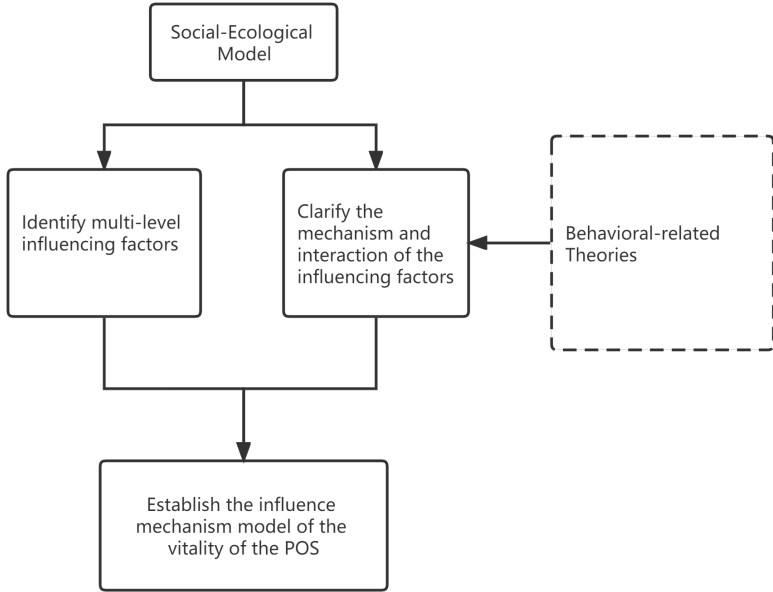

**Figure 3.** Combination of SEM and behavioral-related theories.

Furthermore, the universal characteristics of the model make it impossible to propose clear research hypotheses and intervention strategies for specific problems. Therefore, in a specific context, research must be combined with empirical studies. Nevertheless, the model remains a robust framework for studying behavioral activities, both comprehensive and systematic.

### 4.4. Strategies to Enhance Vitality of POS

Combined with the above analysis, the following suggestions are made to improve the vitality of POSs. First of all, users' requirements for POSs are stratified, so POSs' optimization and promotion strategies should be developed step by step, from the most basic needs, level by level. For example, the government constantly invests resources to optimize the landscape and facilities of POSs; however, the security of users of POSs cannot be guaranteed, so the vitality of POSs will be low. By proposing the hierarchical structure of walking needs, Alfonzo [91] stratified the factors affecting users' walking and improved the optimization scheme with the street walking rate from the most basic and urgent needs. Secondly, the neglect of individual characteristics is often the essential reason for the unfairness of public service. Therefore, enhancing the vitality of POSs should respond to the differences in individual characteristics, including measures for all groups and only for individuals, to form both a universal and targeted measure system. NAZARI [92] proposed to focus on the needs of special groups such as disabled and veterans, to improve the applicability of POSs. Thirdly, behavior influence is a highly integrated systemic project, and the relationship between the influencing factors is complex and intertwined [27]. The ability of each factor to influence behavior has its strengths or weaknesses, and the effective long-term enhancement of the vitality of POSs still needs to be achieved based on measures at one level. Through the collaboration among social, physical, and policy environments, multiple layers of regulation and multidimensional synergy can be achieved to complement, expand, and extend the capabilities of each. For example, the government's guiding policies will influence the residents' attitudes and habits, and the residents' social relationships will influence the effective implementation of policies. From a social and ecological perspective, Tehrani et al. [37] integrated the factors affecting women's physical activity and developed comprehensive interventions.

### 4.5. Identified Gaps in the Literature

Although researchers have analyzed the influencing factors at various levels, studies still need to be conducted on the interactions between the factors and the magnitude of their effects among the various levels. Further quantitative analysis is needed for each factor's weight, correlation, and significance, to understand more clearly the relationship and magnitude of each level and to provide more targeted suggestions and references for improving the vitality of POSs.

In addition, further research is needed to assess the impact of the vitality of POSs in rural, remote areas. This will require careful consideration of appropriate and sophisticated measurement tools to explain the interrelationships between the above-mentioned factors. Studies using SEM may help to better understand how the different factors interact with each other and suggest sustainable and effective strategies for optimal enhancement. In the future, to further verify and confirm the significance and interrelationship of these factors, this study suggests that more empirical quantitative studies are needed.

POSs are the primary vehicles of outdoor activity and socialization for residents, and high-quality POSs play a critical role in promoting individual well-being and provide positive value to the society, economy, and environment of our cities [3]. Improving the vitality of POSs and making full use of existing spaces are necessary for the sustainable development of POSs as well as for public well-being and health. This study uses the SEM as a theoretical framework to address the problem that the vitality of some POSs is lower than expected, and to conduct a comprehensive and systematic investigation of the factors that influence the vitality of POSs. Using the PRISMA method, 34 articles in

this field were systematically reviewed, and the SEM framework was developed in the context of user behavior for POS vitality. The determinants of the vitality of POSs were identified at four levels: individual characteristics of users, social environment, physical environment of the space, and political environment. Criticizing the problems of the existing literature, it emphasizes the influence of political environment factors on the vitality of POSs, negates the improvement of physical environment as a panacea, and blindly focuses on the optimization of the physical environment. A model for the future development of SEM is also proposed.

The SEM broke through the limitations of the theoretical model based on one-sided factors and established that it contains individual factors, material environment, social environment, and policy environment factors of comprehensive influence. Thus, the discussion of the behavioral influencing factors in a more comprehensive context provokes the discussion of the influencing factors from dispersion to a comprehensive system. The ultimate goal of exploring the influence mechanism of POS behavior activities is to obtain targeted POS vitality enhancement strategies. The multidimensional interaction among the influencing factors suggested by the SEM is the necessary disclosure of the action mechanism of the influencing factors. A deep understanding of this is conducive to the proposal of precise strategies. Since there is a multidimensional interaction between influencing factors, relevant strategies and measures must start from multiple fields or departments simultaneously to form a mechanism of interaction and cooperation to avoid the absence of some fields or departments.

**Author Contributions:** Writing—original draft preparation, D.Z.; writing—review and editing, G.H.T.L.; writing—review and editing, S.H.b.M.; Searching—identification of studies, M.F. All authors have read and agreed to the published version of the manuscript.

**Funding:** This research received no external funding.

**Institutional Review Board Statement:** Not applicable.

**Informed Consent Statement:** Not applicable.

**Data Availability Statement:** Not applicable.

**Conflicts of Interest:** The authors declare no conflict of interest.

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
