# Peer review of "A Systematic Review of Factors Influencing the Vitality of Public Open Spaces: A Novel Perspective Using Social–Ecological Model (SEM)"

_sustainability, doi:10.3390/su15065235_

Round 1
Reviewer 1 Report
(1) Some terminologies, statements, and numbers need to have clear definitions or references.
(1.1) For example, "The Sustainable Development Goal by 2030" mentioned in the first sentence should have a reference.
(1.2) Since POS is the main theme, it would help if authors made an effort to define it. Is it a legal construct or just a well-accepted definition?
(1.3) SEM should be used to replace Social-Ecological Model in any parts of the manuscripts
(1.4) The citation style needs to be consistent throughout the manuscripts (e.g., see line 292)
(2) Methods section should be revised
(2.1) The methods section only describes the systematic review process. It should have a part describing how SEM was applied – may consider moving some sentences from section 3.1 and the introduction to the methods section.
(2.2) The procedure describes that Zhang and Fang independently did everything (searched for articles, reviewed the titles and abstracts of the articles found, reviewed the full texts, and selected articles for inclusion). What happened if there were disagreements? The writing process needs to have a clear strategy for preventing bias.
(2.3) The inclusion and exclusion criteria are said to be listed in Table 1 but the table only contains inclusion criteria.
(2.4) “We removed 239 duplicates and 146 other irrelevant articles in the screening process” These records were removed before the screening step. Were there any additional exclusion reasons to exclude 515 records at the screening step?
(2.5) There are some asterisks in Figure 1 but they are not explained.
(2.6) Authors may consider using a quality assessment tool to assess each article for quality and bias because PRISMA is not intended to be a quality assessment tool and it should not be used as such. It can help reduce the number of final selected articles by excluding the articles that only satisfy inclusion criteria but have low quality.
(3) The results and discussion were written and organized well with three clear parts: Individual Characteristics, Physical Environment, and Social Environment. However, in the Physical Environment, and Social Environment parts, results from studies were just listed with no or little discussion. Authors should spend an effort to understand the reasoning behind the results. For example, are there any disparities between the results of selected studies (e.g., Liu et al. (2021) found that accessibility can have a negative impact on the use of POS)? If yes, what are the reasons?
Also, authors need to be careful when interpreting the results from studies. Again, take Liu et al. (2021) as an example, traffic accessibility is different from accessibility in general.
(4) Given POS is the main topic, there should be some discussion on the origin of the studies such as regions (North America, Europe, South East Asia, etc.), countries (high-income or low and middle-income countries), city areas (urban, periphery or rural areas). The origin of the studies can link to several factors (e.g., urban development history, urban design styles, local culture, and policies) that influence the formation and use of POS. It could help to explain the similarities and disparities between the results of studies. Also, a potential limitation of this systematic review is that if all or majority of the selected studies come from a certain region, authors wouldn’t be able to make statements about the vitality of POS in general.
(5) Regarding (4), authors may consider having a section to describe the limitations of this systematic review.
(6) Figure 2 and 3 need to be regenerated. They’re not clear with low resolution and have too many fonts and text sizes. It’s not clear where the line arrows in figure 2 come from (from the whole figure on the left or one of its four sections?). Should “Companies activities in POS” has an arrow pointing to it?
(7) Authors may consider adding to Table 2 the location of the studies (see (4)) or sample size (see(2.6)) if applicable.
Author Response
Response to Reviewer 1 Comments
Manuscript ID: sustainability-2250893
Dear Reviewer,
We feel great thanks for your professional review work on our article. As you are concerned, there are several problems that need to be addressed. According to your nice suggestions, we have made extensive corrections to our previous draft, the detailed corrections are listed below. Our response is given in “red color” text, and changes/additions to the manuscript are shown in “blue color” for your ease of reference. (attachment is the new manuscript)
Point 1: Some terminologies, statements, and numbers need to have clear definitions or references. For example, "The Sustainable Development Goal by 2030" mentioned in the first sentence should have a reference.
Response 1: As suggested by the reviewer, we have added reference to support this idea, for more detail see line 24.
Point 2: Since POS is the main theme, it would help if authors made an effort to define it. Is it a legal construct or just a well-accepted definition?
Response 2: We added the definition of POS: POS are defined as open to all peoples, including public assembly places and connecting spaces, such as squares, parks, sidewalks, and streets. These spaces are not limited by factors such as gender, race, age, or socioeconomic level. For more detail see line 24-27.
Point 3: SEM should be used to replace Social-Ecological Model in any parts of the manuscripts
Response 3: We have made the corrections to make the word harmonized within the whole manuscript.
Point 4: The citation style needs to be consistent throughout the manuscripts (e.g., see line 292)
Response4: We have made the corrections to make the citation style harmonized within the whole manuscript.
Point 5: The methods section only describes the systematic review process. It should have a part describing how SEM was applied – may consider moving some sentences from section 3.1 and the introduction to the methods section.
Response 5: We have added a description of how SEM was applied in the methods section.
In reviewing the literature on the influencing factors for the vitality of the POS, this paper will also use SEM as an organizational framework to synthesize studies related to the user behavior impact of POS. The factors affecting the vitality of POS were determined through literature combing. Combined with SEM, the influencing factors were described in different levels, and the factors were summarized into four levels of SEM, which were individual characteristics, social environment, physical environment and policy environment. Following an overview of the literature, we will discuss the understudy of the current literature, the possible development of SEM in the study of influencing factors, and strategies for the POS vitality boost. (See line 137-145)
Point 6: The procedure describes that Zhang and Fang independently did everything (searched for articles, reviewed the titles and abstracts of the articles found, reviewed the full texts, and selected articles for inclusion). What happened if there were disagreements? The writing process needs to have a clear strategy for preventing bias.
Response 6: A third reviewer was used to resolve a disagreement where necessary, and we added a part describing how we screen the articles and prevent bias. (See line 111-120)
Point 7: The inclusion and exclusion criteria are said to be listed in Table 1 but the table only contains inclusion criteria.
Response 7: We have added the exclusion criteria in table 1. (Page 3)
Point 8: “We removed 239 duplicates and 146 other irrelevant articles in the screening process” These records were removed before the screening step. Were there any additional exclusion reasons to exclude 515 records at the screening step?
Response 8: We have added the exclusion criteria in table 1, as well as a part describing how we screen the articles and prevent bias.
All search articles were retrieved and uploaded in the Zotero tool for systematic reviews. The abstract and title of the study, as well as the full text screening were done by two independent reviewers (Zhang, Fang), with a third reviewer used to resolve disagreements.
For data extraction, a standard template was used, containing details of each articles’ title, author, date, title, context, sample size and data collection, influencing factors, and outcome of using POS.
Quality appraisal of studies was conducted by two reviewers using the QUALSYST, as QUALSYST can assess the quality and potential bias in a wide range of study designs, from experimental to observational. All articles were evaluated in five areas: study context, data collection methods, sample size, influencing factors, and results. (See line 111-120)
Point 9: There are some asterisks in Figure 1 but they are not explained.
Response 9: We have deleted the asterisks in Figure 1. (Page 4)
Point 10: Authors may consider using a quality assessment tool to assess each article for quality and bias because PRISMA is not intended to be a quality assessment tool and it should not be used as such. It can help reduce the number of final selected articles by excluding the articles that only satisfy inclusion criteria but have low quality.
Response 10: We added the quality assessment tool, which is QUALSYST. Quality appraisal of studies was conducted by two reviewers using the QUALSYST, as QUALSYST can assess the quality and potential bias in a wide range of study designs, from experimental to observational. All articles were evaluated in five areas: study context, data collection methods, sample size, influencing factors, and results. (See line 117-120)
Point 11: The results and discussion were written and organized well with three clear parts: Individual Characteristics, Physical Environment, and Social Environment. However, in the Physical Environment, and Social Environment parts, results from studies were just listed with no or little discussion. Authors should spend an effort to understand the reasoning behind the results. For example, are there any disparities between the results of selected studies (e.g., Liu et al. (2021) found that accessibility can have a negative impact on the use of POS)? If yes, what are the reasons?
Also, authors need to be careful when interpreting the results from studies. Again, take Liu et al. (2021) as an example, traffic accessibility is different from accessibility in general.
Response 11: We have added the discussion of Social Environment. The social environment can reflect the interpersonal relationship or cultural atmosphere in a specific place and will affect the specific behavior of POS users. Behavior-related theories often incorporate social and environmental factors into constructing behavioral decision-making relationships. The literature review summarizes the factors of social environment into the following two points: social support and the sense of belonging. Then we discussed the social environment in these two points. (See line 235-255)
We also added the discussion of physical environment. Starting from the actual needs of users, we discussed that the improvement of the material environment should be divided in different levels. We need to start from the most basic needs of users, otherwise it cannot bring ideal results. (See line 288-297).
Point 12: Given POS is the main topic, there should be some discussion on the origin of the studies such as regions (North America, Europe, South East Asia, etc.), countries (high-income or low and middle-income countries), city areas (urban, periphery or rural areas). The origin of the studies can link to several factors (e.g., urban development history, urban design styles, local culture, and policies) that influence the formation and use of POS. It could help to explain the similarities and disparities between the results of studies. Also, a potential limitation of this systematic review is that if all or majority of the selected studies come from a certain region, authors wouldn’t be able to make statements about the vitality of POS in general.
Response 12: We add to Table 2 the context of POS, sample size and data collection. Meanwhile, we added a section to describe the limitation of this systematic review and reliability of result. We explored the limitations of this literature review from the context of the POS in the selected articles. We also explained the credibility of the literature review conclusions from the number of samples. (See line 323-334)
Point 13: Regarding (4), authors may consider having a section to describe the limitations of this systematic review.
Response 13: As suggested, we added a section to describe the limitation of this systematic review. We explored the limitations of this literature review from the context of the POS in the selected articles. (See line 326-337).
Point 14: Figure 2 and 3 need to be regenerated. They’re not clear with low resolution and have too many fonts and text sizes. It’s not clear where the line arrows in figure 2 come from (from the whole figure on the left or one of its four sections?). Should “Companies activities in POS” has an arrow pointing to it?
Response 14: We have regenerated the figure 2 and figure 3, and to clear the relationship in figure 3, we changed the description and figure 3. (See line 394-411)
Point 15: Authors may consider adding to Table 2 the location of the studies (see (4)) or sample size (see (2.6)) if applicable.
Response 15: We have added the context of POS and sample size in table 2.
We tried our best to improve the manuscript and made some changes marked in blue highlight text in revised paper which will not influence the content and framework of the paper. We appreciate for your warm work earnestly, and hope the correction will meet with approval. Once again, thank you very much for your comments and suggestions.
Best Regard
Zhang Danning

Reviewer 2 Report
The paper presents a systematic literature search using the PRISMA method to screen and select articles from three major databases (Science Web, Elsevier, and Scopus), tackling the topic of spatial planning, architectural design, and management of Public Open Spaces (POS). The framework examines the influence of several factors classified into four fields: individual user characteristics, social environment, physical environment, and political environment. Furthermore, the contribution provides a theoretical contribution by conceptualizing the Social-Ecological Model (SEM) framework, summarizing the influencing factors at different levels, and providing a critical analysis of the strengths and weaknesses of the SEM.
The topic is very interesting and could effectively support the main stakeholders involved in the decision-making process for transforming urban areas to upgrade their urban quality and improve the vitality of POS.
However, I think the paper should be revisited to make some parts clearer and more complete.
Please find some comments and remarks below:
Abstract:
- Line 14-16: “The study of these factors should also have practical implications, as the results will ultimately provide improvement strategies to help policymakers and local communities effectively and sustainably improve the vitality of POS.” > This part should be further developed. A final paragraph with practical examples could be very useful in order to better understand the approach that should be adopted for improving the vitality of POS.
1. Introduction
- Line 31-32: “Little attention has been paid to the problem of underutilization of POS.” > This statement should be supported by statistics data and/or references in literature.
- Line 35-36: “The number of users can judge the performance of POS.” > It is maybe better to use “represent a significant indicator for the” instead of “judge”.
- Line 36-37: “The more and the more characteristic the visitors are, the better the quality of POS [11].” > In which sense? Please specify better the term “characteristic”.
- Line 70: “physical activity behavior” > Please specify this expression better.
2. Methods
- Line 92: “included” > It is better to replace this adjective with a noun, e.g. “inclusiveness” or “inclusion”.
- Line 99-101: “In addition, articles included in this review had to meet the inclusion and exclusion criteria listed in Table 1.” > Actually, in Table 1, the exclusion criteria are not listed, so it is better to say “to meet the inclusion criteria listed in Table 1.”
- Lines 102-113: This paragraph is not totally clear and should be more detailed in relation to data reported in Figure 1.
In detail:
- Line 105: “We then reviewed the titles and abstracts of the articles” > How many articles were finally evaluated? Please specify the number (such as “the remaining/selected 585 articles”).
- Line 108: “In the next step, we further evaluated the articles” > How many articles were finally evaluated? Again, please specify the number.
- Figure 1: Add a note for “***” after “Records excluded” in order to explain why n = 515 articles have been excluded.
3. Results and discussion
- I suggest reorganizing Chapter “3. Results and discussion” as follows:
Eliminate the title "General Description"; Entitle the 3.2. sub-chapter as “3.1 Individual Characteristics", .... and so on.... until the last sub-chapter "3.4. Political Environment".
Create a new Chapter "4. Title" for the former sub-chapter 3.6-3.10. For this chapter, what is the common thread for the topics tackled in 3.6-3.10? Maybe a critical analysis of the SEM? This is not so clear and should be better detailed.
Therefore, the Chapter "Conclusion" would become "5. Conclusion".
- Lines 122-125: Standardize the way numerical data are reported (by always using numbers or letters).
- Line 139: Add a final “r” to the word “behavior”.
- Line 176: “argued that different incomes have different needs POS.” > Please specify this statement better.
- Lines 180-182: “Marital status also affects the usage patterns of POS residents [53,63,66]. Moreover, the health status of residents affects the utilization behavior of POS [67].”> Specify if these factors influence by increasing or decreasing the need for POS and their utilization.
- Figure 2: The relation between the four dimensions and the outcomes is not clear.
In the flowchart, are the arrows of the outcomes related to the four different dimensions or not?
- Sub-chapter “3.7 Physical environment improvement as a panacea”: Please specify better if "physical environment improvement" stands for the increase of pleasantness or aesthetic-formal architectural quality or other. Physical refers more to the technical parameters linked to the physics sphere, so I suggest using another term (such as "pleasantness" or "beauty"). For this part, I think it is crucial to refer to the current New European Bauhaus (NEB) values launched by the European Union for promoting the renovation of living spaces according to the principles of beauty, sustainability and inclusiveness ("beautiful, sustainable, together").
- Sub-chapter “3.8. Future development of SEM”: It is not clear which improvement is suggested by the authors in order to implement the SEM. Please specify better the part related to "Behavioral decision analysis" included in the Figure 3 chart.
General recommendations:
- Once introduced the definition of “Social-Ecological Model (SEM)” (in line 54), always use the acronym SEM throughout the text (e.g. line 83, Table 1, first row of Table 2, line 107, line 256, caption of Figures 2 and 3, line 308, lines 371-372, line 383, line 390, …).
- Once introduced the definition of “Public Open Spaces (POS) (in line 2 of the Abstract), always use the acronym POS throughout the text (e.g. line 107, …).
- Finally, I think it would also be important to take a look at research projects at the European level that are working on developing tools and methods for evaluating living spaces (for example, under the NEB initiatives: NetZeroCities https://netzerocities.eu and CrAFt https://craft-cities.eu).
Author Response
Response to Reviewer 2 Comments
Manuscript ID: sustainability-2250893
Dear Reviewer,
We sincerely thank the editor and all reviewers for their valuable feedback that we have used to improve the quality of our manuscript. Our response is given in “red color” text, and changes/additions to the manuscript are shown in “blue color” for your ease of reference.
Point 1: Line 14-16: “The study of these factors should also have practical implications, as the results will ultimately provide improvement strategies to help policymakers and local communities effectively and sustainably improve the vitality of POS.” > This part should be further developed. A final paragraph with practical examples could be very useful in order to better understand the approach that should be adopted for improving the vitality of POS.
Response 1: We have added the specific strategies in abstract (see line 12-16) and the practical examples in line 421-444.
- Introduction
Point 2: Line 31-32: “Little attention has been paid to the problem of underutilization of POS.” > This statement should be supported by statistics data and/or references in literature.
Response 2: We have added reference to support this idea, for more detail see line 39.
Point 3: Line 35-36: “The number of users can judge the performance of POS.” > It is maybe better to use “represent a significant indicator for the” instead of “judge”.
Response 3: As suggested by the reviewer, we have changed this sentence. “The number of users can represent a significant indicator of the performance of POS.” (See line 42-43)
Point 4: Line 36-37: “The more and the more characteristic the visitors are, the better the quality of POS [11].” > In which sense? Please specify better the term “characteristic”.
Response 4: We have changed the phraseology: “The more the number and type of visitors, the better the quality of POS and the more the public space can meet the needs of different user groups.” (See line 43-45)
Point 5: Line 70: “physical activity behavior” > Please specify this expression better.
Response 5: We have given the example of “physical activity behavior”. (See line 80)
- Methods
Point 6: Line 92: “included” > It is better to replace this adjective with a noun, e.g., “inclusiveness” or “inclusion”.
Response 6: According to the suggestion, we have changed to “inclusion”. (See line 101)
Point 7: Line 99-101: “In addition, articles included in this review had to meet the inclusion and exclusion criteria listed in Table 1.” > Actually, in Table 1, the exclusion criteria are not listed, so it is better to say “to meet the inclusion criteria listed in Table 1.”
Response 7: We have added the exclusion criteria in table 1. (Page 3)
Point 8: Lines 102-113: This paragraph is not totally clear and should be more detailed in relation to data reported in Figure 1.
In detail:
- Line 105: “We then reviewed the titles and abstracts of the articles” > How many articles were finally evaluated? Please specify the number (such as “the remaining/selected 585 articles”).
- Line 108: “In the next step, we further evaluated the articles” > How many articles were finally evaluated? Again, please specify the number.
- Figure 1: Add a note for “***” after “Records excluded” in order to explain why n = 515 articles have been excluded.
Response 8: We have rewritten this part according to the suggestion. We added the exact number of articles, and the detail process for articles selection. (See line 111-128)
- Results and discussion
Point 9: I suggest reorganizing Chapter “3. Results and discussion” as follows:
Eliminate the title "General Description"; Entitle the 3.2. sub-chapter as “3.1 Individual Characteristics", .... and so on.... until the last sub-chapter "3.4. Political Environment".
Create a new Chapter "4. Title" for the former sub-chapter 3.6-3.10. For this chapter, what is the common thread for the topics tackled in 3.6-3.10? Maybe a critical analysis of the SEM? This is not so clear and should be better detailed.
Therefore, the Chapter "Conclusion" would become "5. Conclusion".
Response 9: As the suggestion, we have reorganized chapters as follows:
- Introduction
- Methods
- Results and Discussion
- Conclusion and Recommendations
Point 10: Lines 122-125: Standardize the way numerical data are reported (by always using numbers or letters).
Response 10: We have made the corrections to make the numerical data harmonized within the whole manuscript. (See line 153-158)
Point 11: Line 139: Add a final “r” to the word “behavior”.
Response 11: We were really sorry for our careless mistake, thank you for your reminder. (See line 171)
Point 12: Line 176: “argued that different incomes have different needs POS.” > Please specify this statement better.
Response 12: We have specified this statement. Compared with the middle and high-income groups, the low-income groups significantly prefer the needs of sports, social interaction, public participation, and consumption. High-income groups relatively prefer the need for safety and accessibility. (See line 210-213)
Point 13: Lines 180-182: “Marital status also affects the usage patterns of POS residents [53,63,66]. Moreover, the health status of residents affects the utilization behavior of POS [67].”> Specify if these factors influence by increasing or decreasing the need for POS and their utilization.
Response 13: We have specified this statement. Marriage or partnership can be considered a symbol of love and companionship. When divorced, separated, or widowed, it is natural to assume that these people become lonely and unhappy. Therefore, green space, as a place to help reduce loneliness and improve social cohesion, will naturally be attractive to these people and thus become important. (See line 217-221)
Point 14: Figure 2: The relation between the four dimensions and the outcomes is not clear.
In the flowchart, are the arrows of the outcomes related to the four different dimensions or not?
Response14: We have modified the Figure 2. (See page 13)
Point 15: Sub-chapter “3.7 Physical environment improvement as a panacea”: Please specify better if "physical environment improvement" stands for the increase of pleasantness or aesthetic-formal architectural quality or other. Physical refers more to the technical parameters linked to the physics sphere, so I suggest using another term (such as "pleasantness" or "beauty"). For this part, I think it is crucial to refer to the current New European Bauhaus (NEB) values launched by the European Union for promoting the renovation of living spaces according to the principles of beauty, sustainability and inclusiveness ("beautiful, sustainable, together").
Response 15: We have specified the physical environment of POS as landscape of POS, facilities, accessibility etc. (see line 365-367)
Point 16: Sub-chapter “3.8. Future development of SEM”: It is not clear which improvement is suggested by the authors in order to implement the SEM. Please specify better the part related to "Behavioral decision analysis" included in the Figure 3 chart.
Response 16: We have rewritten this part and redrew figure 3 according to the suggestion. In the future study of the vitality of POS, SEM can be combined with the behavioral-related theories, use SEM to summarize the influencing factors, use the behavioral-related theories to clarify the mechanism and interaction of influencing factors, and finally establish a model of the vitality influence mechanism of POS. (See Line 394-411)
General recommendations:
Point 17: Once introduced the definition of “Social-Ecological Model (SEM)” (in line 54), always use the acronym SEM throughout the text (e.g. line 83, Table 1, first row of Table 2, line 107, line 256, caption of Figures 2 and 3, line 308, lines 371-372, line 383, line 390, …).
Response 17: We have made the corrections to make the word harmonized within the whole manuscript.
Point 18: Once introduced the definition of “Public Open Spaces (POS) (in line 2 of the Abstract), always use the acronym POS throughout the text (e.g. line 107, …).
Response 18: We have made the corrections to make the word harmonized within the whole manuscript.
Point 19: Finally, I think it would also be important to take a look at research projects at the European level that are working on developing tools and methods for evaluating living spaces (for example, under the NEB initiatives: NetZeroCities https://netzerocities.eu and CrAFt https://craft-cities.eu).
Response 19: Thanks for your suggestions. This study mainly focuses on applying SEM to generalize the four levels of factors that affect the vitality of POS. Suppose we improve the landscape of POS, such as accessibility, comfort, and vitality of POS. In that case, we are also close to the beauty and sustainable inclusiveness proposed by New European Bauhaus.
We tried our best to improve the manuscript and made some changes marked in blue highlight text in revised paper which will not influence the content and framework of the paper. We appreciate for your warm work earnestly, and hope the correction will meet with approval. Once again, thank you very much for your comments and suggestions.
Best Regards
Zhang Danning

Round 2
Reviewer 2 Report
The authors significantly revised the manuscript, changing the structure and implementing the contents as suggested.